# GENERATIVE AUTO-ENCODER: CONTROLLABLE SYNTHESIS WITH DISENTANGLED EXPLORATION

## ABSTRACT

Autoencoders are a powerful information compression framework, but it lacks generative ability. We explore whether an autoencoder can gain generative ability without invoking GAN-based adversarial training, which is notoriously hard to train and control. VAE provides one solution by adding Gaussian prior knowledge that can be sampled for synthesis but faces a low-quality problem. One naive way is directly through exploration in latent space such as interpolation; however, randomly interpolating the latent code may wander out of the acceptable distribution of the decoder, which leads to inferior output. Here we propose a new method: Disentangled Exploration Autoencoder (DEAE), which uses disentangled representation and regularization to guarantee the validity of exploration in latent space and achieve controllable synthesis. The encoder of DEAE first turns the input sample into a disentangled latent code, then explores the latent code space through directed interpolation. To aid the interpolated latent code in successfully outputting a meaningful sample, after the decoder, we regularize the output by 'reusing' the encoder to force the obtained latent representation to maintain perfect disentanglement, which implicitly improves the quality of the interpolated sample. The disentanglement and exploration can boost each other and form a positive loop that empowers DEAE's generative ability. Experiments demonstrate that DEAE can improve the performance of downstream tasks by synthesizing attribute-controllable augmented samples. We also demonstrate that DEAE can help to eliminate dataset bias, which provides a solution for fairness problems.

## 1 INTRODUCTION AND RELATED WORK

Autoencoders (Ballard (1987)) usually map the samples from a high dimension space to a latent, low-dimensional space with minimum information loss and employ regularization function to help downstream tasks. The compressed low-dimensional space provides a high-level and compact representation to help understand the original dataset (Berthelot et al. (2018)). It is interesting to analyze the relationship between the latent space and image space (Bengio et al. (2013)), especially how to empower the autoencoder to obtain generative ability. Prior work exists which turns an autoencoder into a generative model. These methods can be divided into two types: Adversarial training and VAEs. As is shown in Fig. 1 (a), Makhzani et al. (2015); Zhao et al. (2018) uses adversarial learning in latent space to control the output of encoder which have similar distribution as the real data. Berthelot et al. (2018) uses an adversarial regularizer to improve interpolation in autoencoders. Other methods Sainburg et al. (2018) merge adversarial learning and autoencoder by adding a discriminator to guide the quality of synthesized samples. However, adversarial-based learning suffers from unstable training and synthesis that is hard to control (Gulrajani et al. (2017)). As is shown in Fig. 1 (b), VAEs (Kingma & Welling (2014); Higgins et al. (2017); Chen et al. (2018)), provide another solution by adding prior Gaussian constraints in latent space, then using latent code sampling to synthesize new samples. Zhang et al. (2019) add latent distribution constraint, which yields sharper outputs. However, VAEs suffer from low-quality problems and convergence difficulty.

Moreover, both types of methods struggle to achieve controllable semantic synthesis. Adversarial-based controllable synthesis is mostly achieved by interpreting the latent space by finding the boundary (Shen et al. (2020a); Yang et al. (2019)), which has a high cost; most of the controllable synthesis is then restricted to binary attribute values and one cannot precisely control the synthesized image without influencing the other attributes. VAE-based controllable synthesis achieved by disentangling

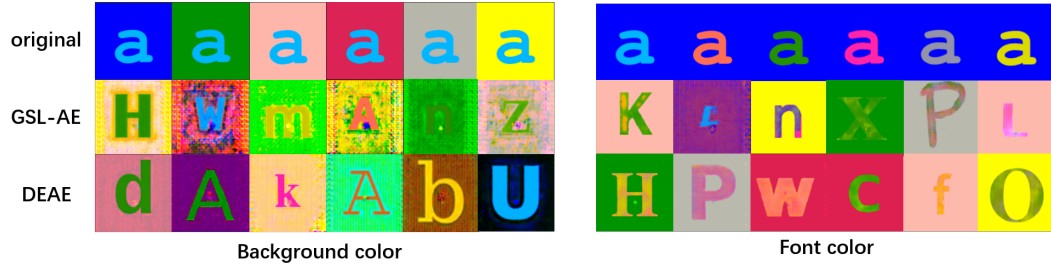

(a) Adversarial Autoencoder  (b) $\beta$-VAE  (c) DEAE

Figure 1: Different methods for empowering autoencoder generative ability

the representation in latent space struggles with the problems of hard to control and inferior synthesis quality. We propose a different solution to empower precise attribute controllable synthesis ability on autoencoders: Disentangled Exploration Autoencoder (DEAE).

Direct exploration in latent space, such as interpolation, is a naive way for autoencoders to generate new samples. Hopefully, the synthesized samples by the decoder can achieve a semantically meaningful output. However, there are two obstacles to directly interpolating without constraints. First, the interpolated latent code may get beyond the boundary of the non-convex distribution that the pre-trained decoder can 'understand', especially when the training dataset is small. Second, even though the synthesized samples may have high quality, it's still hard to precisely control the synthesized samples to change only the target attribute without influencing other attributes.

To solve the problem of interpolation and aid semantic controllable synthesis in autoencoders, DEAE (Fig. 1 (c)) first turns the latent code into an attribute disentangled representation after encoding, through disentangled representation learning (Ge et al. (2020)). It can then directly interpolate along any dimension of the disentangled latent code to change specific semantic attributes of output samples. For example, if object color is specified in the first 3 dimensions of the latent code, exploring new object colors is achieved by interpolating within these 3 dimensions. But how to make sure the synthesized samples are semantically meaningful? We propose to 'reuse' the encoder as a regularization, with the assumption that if the synthesized interpolated sample is semantically meaningful, it should be properly encoded back to the latent space by the encoder. Furthermore, because we only interpolate a specific attribute in disentangled latent code, when we encode the decoded synthesized sample back to latent code again, the resulting latent code for all other attributes should be the same as the original. Thus, our regularization procedure can 'reuse' the encoder and constrain the latent values, which implicitly improves the quality of the interpolated samples and achieves precise attribute-controllable synthesis.

More importantly, the disentangled representation and exploration procedure can help each other during training. On the one hand, the disentangled representation in latent code makes the attribute-controllable exploration possible, and it is also the foundation of the regularization by reusing the encoder. On the other hand, the better the quality of interpolated samples, the better the disentanglement by the encoder. This positive loop is crucial for our DEAE training. Compared with other GAN-based controllable synthesis methods like InterFaceGAN (Shen et al. (2020a)), Fader Networks (Lample et al. (2018)), and starGAN (Choi et al. (2018)), which can be treated as the first exploration then control, our method reverses the procedure by first control then exploration.

Our contributions are: (i) Propose Disentangled Exploration Autoencoder (DEAE), which uses disentangled representation and interpolation to boost each other in latent space and achieve the interpolation-based attribute-controllable synthesis, including synthesis for novel attribute values

Figure 2: Controllable mining novel background and font color by interpolation in latent space.

never seen in training. (ii) Demonstrate how DEAE can use exploration to improve the comprehension of the bridge between disentangled latent space and original space, where precise movement in latent space results in changing specific attribute(s) to a specific target value(s). (iii) Demonstrate that DEAE outperforms state-of-the-art alternatives for controllable image synthesis and can be used as a data augmentation method which can boost other downstream tasks. (iv) Demonstrate that DEAE can help eliminate dataset bias and improve the performance of downstream tasks.

## 2 DISENTANGLED EXPLORATION AUTOENCODER FOR ATTRIBUTE CONTROLLABLE SYNTHESIS

An autoencoder consists of two main structures: first an 'encoder', which takes $x \in \mathbb{R}^{d_x}$ as input and obtains a latent code $z = f_\theta(x)$, $z \in \mathbb{R}^{d_z}$. Then a 'decoder', which maps the latent code to an approximate reconstruction $\hat{x} = g_\phi(z)$ of the input $x$. In this paper, we consider the situation that both encoder and decoder are neural networks with trainable parameters $\theta$ and $\phi$ respectively. Given a multi-attribute dataset $D$, each sample $x \in D$ has m attributes and each attribute has different attribute values. Take the dSprites dataset Matthey et al. (2017) as an instance, color, shape, scale, orientation, and X, Y positions are different attributes; the shape attribute, for example, has three values: square, ellipse, and heart. Our goal is to controllable synthesize new samples on specific attribute(s). For example, given one image, synthesize new images with different shapes while the other five attributes remain unmodified. The challenging part is to constrain the other attributes to remain unmodified when modifying the target attribute.

DEAE uses the following steps to achieve controllable synthesis. First, use the Group supervised learning Ge et al. (2020) method to create a preliminary disentangled representation in latent space, where different attributes are disentangled with corresponding latent dimensions (Sec. 2.1). Then, explore the disentangled latent space by interpolating and synthesizing. Then we 'reuse' the encoder to regularize the synthesized samples, which makes the disentangled representation and exploration help each other (Sec. 2.2). This positive loop helps creating a *perfect* disentangled autoencoder, where controllable synthesis can be achieved by freely interpolating in latent space (Sec. 2.3). Through perfect disentanglement, DEAE creates a subspace $\mathbb{R}^{d_{z_i}}$ for each attribute $a_i$ in latent code, which provides a useful view to exploring how to precisely control synthesis for each attribute (Sec. 2.4).

### 2.1 PRELIMINARY DISENTANGLED AUTOENCODER BY GSL

We use the Group supervised learning framework Ge et al. (2020) to train an autoencoder with disentangled representation in latent space as our baseline model (GSL-AE). Here are the steps: First, we organize the original dataset as a MultiGraph, where each sample is represented as a $node$, and we use $edges$ to represent the attribute relationships between nodes: if two nodes have the same attribute, e.g., same object color, they are linked by an edge. There may be multiple edges between two nodes. Second, we use the MultiGraph to guide $group$ supervised training: at each training step, we sample a group of samples which is a sub-MultiGraph from the whole MultiGraph as input, swap the latent code of the overlap attribute(s) as long as there are edges between them. We use attribute swap loss and cycle swap loss (Ge et al. (2020)) to mine samples with attributes similar to those in the group. This achieves disentangled representation for the given dataset. This disentanglement can synthesize new samples by recombining latent codes from different source samples.

### 2.2 EXPLORATION BY INTERPOLATION AND DISENTANGLEMENT REGULARIZATION

The generative ability of GSL-AE is powerful, however, there are limitations because the decoder can only 'understand' seen attribute values: the recombined latent code does represent a new combination but each attribute value related dimensions, as a unit, is seen in other dataset samples. This limitation hinders interpolation-based synthesis because each interpolation in latent code can be treated as an 'unseen' attribute value, and the decoder may not be able to 'understand' this new attribute and turn it back to a new sample with proper semantic meaning. For example, in Fig. 3 (a), we want to change the background color of an input image by interpolating the background-color-related dimensions (20 dim) in disentangled latent code. If the latent code is non-convex, during interpolation from the source sample (red point) to the target sample (green point), the middle point may out of the boundary, which means the decoder might not successfully turn it onto a meaningful image with the requested background color. The result of GSL-AE in Fig. 2 illustrates this problem.

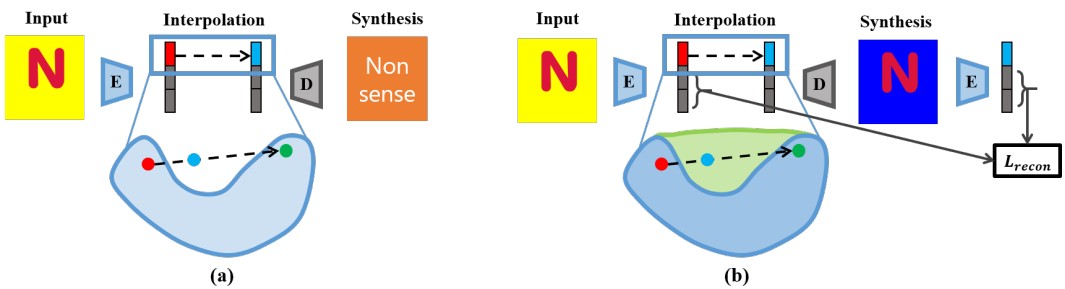

Figure 3: Enlarging explainable region by interpolation and regularization

To solve the problem of non-convex latent space, we define a *perfect* disentanglement property in latent $d_z$ dimension space of the autoencoder. For encoder $f_{\theta*}$, if we specifically modify one attribute value of image $x$ and get $\hat{x}$, and after encoding we get latent code $z = f_{\theta*}(x)$ and $\hat{z} = f_{\theta*}(\hat{x})$, then the difference between $z$ and $\hat{z}$ should be zero for all latent dimensions except those that represent the modified attribute. Similarly, for decoder $g_{\phi*}$, the change in latent space should only influence the corresponding attribute expression in image space. If both the encoder and decoder satisfy the requirements, we claim the autoencoder has a *perfect* disentanglement property. During training, we augment the original training set by randomly interpolating new attribute values that lie between the values of the training set. The constraints of perfect disentanglement then enforce that the latent representation of every attribute is understandable by the decoder, i.e., it forces, in the limit of infinite interpolated samples, the disentangled latent representation of every attribute to be *convex*.

DEAE tries to turn the non-convex latent space to convex for each attribute by 'reusing' the encoder to regularize the latent space of synthesized images. As shown in Fig. 3 (b), after we pass the synthesized interpolated image through the encoder again and get the latent code, we force it to be the same as the latent code of the input image, except for the interpolated attribute related range, by adding a reconstruction loss. This disentangled regularization helps the autoencoder 'complete' the understandable field of each disentangled latent space and turn the original non-convex space to a convex space to achieve perfect disentanglement. If it is possible that the reconstruction loss decreases to zero for a given dataset augmented by many interpolated samples, then perfect disentanglement and convexification are achieved. After that, we can freely interpolate any specific attribute related dimensions to achieve controllable synthesis. Fig. 2 DEAE results show the superiority of DEAE based on the proposed methods over other models.

The exploration of the latent space and disentangled representation of attributes in DEAE are dual tasks in the sense that one can help enhance the performance of the other.

### 2.3 TOWARDS CONTROLLABLE EXPLORATION DIRECTION

After obtaining the perfect disentangled convex latent space for each attribute, in addition to synthesizing new samples by random interpolation, we want to find some more precise and effective methods for attribute-controllable synthesis problems. Here we discuss two ways: precise target synthesis, and mining new attribute values.

**Precise attribute synthesis** means we want to change a specific attribute from value A to value B with the desired trajectory. If we treat the precise attribute synthesis as a 'movement' in attribute-related latent space, moving with the desired trajectory means we can control the speed of movement (most effective path) and the direction during movement.

**Novel attribute value mining** means finding new attribute values that we have never seen before during training. For instance, given only red, green, and blue object color images, can we synthesize new object colors such as purple, yellow, or any other colors? Can we find an effective method that gives the highest probability to find new attribute values during exploration in latent space? To solve the problems above, we put forward the unit direction vector (UDV) for each attribute. In detail, we treat each attribute value as a binary semantic label (e.g., car or not car). We assume there exists a hyperplane in the latent space serving as the separation boundary (Shen et al. (2020b)), and the distance from a sample to this hyperplane is in direct proportion to its semantic score. So, we can train an SVM to find this boundary and use the vector orthogonal to the boundary and towards the

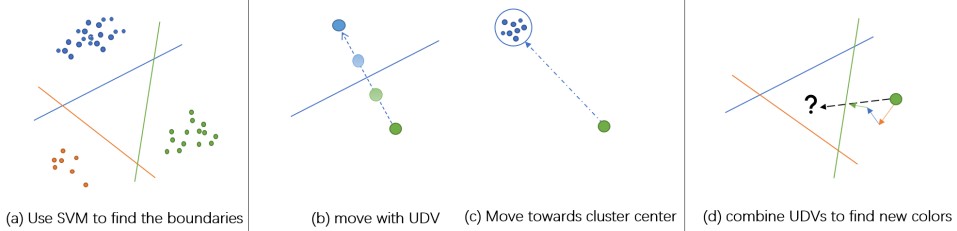

(a) Use SVM to find the boundaries    (b) move with UDV    (c) Move towards cluster center    (d) combine UDVs to find new colors

Figure 4: Towards controllable exploration direction

positive side to represent the UDV. Then we can use the UDVs or the combination of them to achieve precise attribute synthesis and to find new attribute values. Controlling attributes with UDVs is a general method which can be used in most cases. However, for those attributes (object color) whose attribute value (blue color) has small intra-class variance (all blues look similar), their distribution can be seen as a Gaussian distribution, and we can use K-means (Likas et al. (2003)) to find the center of each object color and compute the mean of latent value for those cluster centers.

As is shown in Fig. 4 (a) we can find the boundaries and UDVs by SVM for each attribute value. Then we can solve the precise attribute synthesis problem, as is shown in Fig. 4 (b), by moving from the start point, across the boundary, to the target attribute value, by adding the UDV of the target attribute step by step. Fig. 4 (c) illustrates that we can move the attribute value to the z value of the cluster center directly if the distribution of attribute is Gaussian. Fig. 4 (d) shows that we can combine the UDVs to discover new attribute values.

## 2.4 ARCHITECTURE AND OBJECTIVE FUNCTION

We use the same architecture of autoencoders as Ge et al. (2020). The encoder $E$ is composed of two convolutional layers with stride 2, followed by 3 residual blocks, followed by a convolutional layer with stride 2, followed by reshaping the response map to a vector, and finally two fully-connected layers to output 100-dim vector as the latent feature. The decoder $D$, symmetric to $E$, take the 100-dim vector as input and output a synthesized samples. Besides the reconstruction loss $L_r$, swap reconstruction loss $L_{sr}$ and cycle swap reconstruction loss $L_{csr}$ which are same as GSL, we add a regularization reconstruction loss $L_{reg}$.

$$z = f_\theta(x); \quad z^{re} = f_\theta(g_\phi(z_{ai})) \tag{1}$$

where $z_{ai}$ demotes the latent vector after we interpolate in attribute $ai$ related dimension in latent vector $z$. $z^{re}$ denotes the latent space after reusing the encoder to the synthesized image.

$$L_{reg} = ||z_{-ai} - z^{re}_{-ai}||_{l1} \tag{2}$$

$z_{-ai}$ and $z^{re}_{-ai}$ denote the all latent dimensions of $z$ and $z^{re}$ respectively, except those that represent the modified attribute $ai$

The total loss function is:

$$\mathcal{L}(E, D) = L_r + \lambda_{sr}L_{sr} + \lambda_{csr}L_{csr} + \lambda_{reg}L_{reg}, \tag{3}$$

Where $\lambda_{sr}, \lambda_{csr}, \lambda_{reg} > 0$ control the relative importance of the loss terms.

## 3 EXPERIMENTS

We conduct 5 main experiments: (3.1) The performance of DEAE compared with baseline models on attribute controllable and interpolation-based random synthesis task with different datasets. (3.2) Demonstrate how disentangled representation and exploration boost each other in DEAE with qualitative and quantitative experiments (3.3) Demonstrate the performance of DEAE to find the direction towards a attribute value with UDV which achieves precise attribute synthesis, and how to combine UDVs targeting for mining new attribute values. (3.4) Demonstrate the performance of DEAE, as a generative model, to controllable augment small datasets and boost the performance of

Figure 5: Performance comparison of interpolation-based synthesis

downstream tasks, such as classification. (3.5) Demonstrate DEAE can use disentangled representation to eliminate the bias in a dataset and improve the performance of downstream tasks, which provides a solution for fairness decision problems. We use the following datasets to explore our topic.

**Fonts**[1]: is a computer-generated RGB image dataset. Each image contains an alphabet letter rendered using 5 independent generating attributes: letter identity, size, font color, background color and font. **RaFD**(Langner et al. (2010)): contains pictures of 67 models displaying 8 emotional expressions taken at 5 different camera angles simultaneously. There are 3 attributes: identity, camera position (pose), and expression.

### 3.1 CONTROLLABLE SYNTHESIS PERFORMANCE

**Attribute controllable synthesis task** On the Fonts dataset, we train our DEAE and three baseline models: General Autoencoder (GAE), $\beta$-Variational Autoencoder ($\beta$-VAE), and Autoencoder with Group supervised learning(GSL-AE) (Ge et al. (2020)).

Fig. 5 (a) shows the performance of attribute controllable synthesis on the background color, size ,and fonts given two images. The first three rows show synthesized images by randomly interpolating background color related dimensions in two samples. $\beta$-VAE can hardly synthesize images which differ only in background color. The synthesized images by GSL also change the size and font range while DEAE can synthesize high quality images that change only the background color attribute while keeping the other attributes unmodified. Size and fonts show similar performance.

On RaFD dataset, we train DEAE to show the performance of controllable synthesis on identity and expression (Fig. 6). The results show that DEAE can achieve controllable synthesis on the expression as well as the identity attribute, generating new expressions and identities between the two sources.

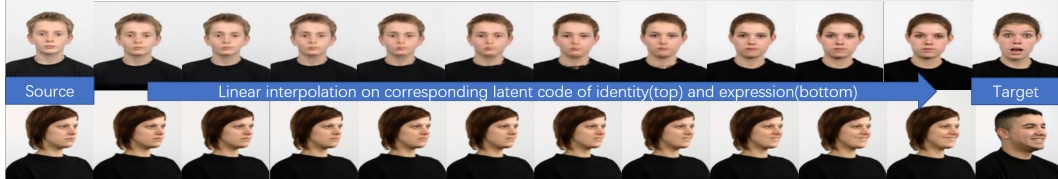

Figure 6: DEAE controllable synthesis performance on RAFD dataset. Linear interpolation on the identity(top) and expression(bottom) attribute between the source image and the target image.

**Interpolation based random synthesis task** As is shown in Fig. 5 (b) we randomly interpolate in latent code for all latent attributes given two samples, the result shows DEAE can synthesize new samples by freely interpolating in latent space.

### 3.2 DISENTANGLEMENT AND EXPLORATION BOOST EACH OTHER

To further explore that DEAE can turn the non-convex space to a convex space by disentangled exploration. We claim that the exploration synthesis and disentangled representation of attributes are dual tasks in the sense that one can help enhance the performance of the other. Intuitively, the

---

[1]http://ilab.usc.edu/datasets/fonts

regularization loss to improve the quality of interpolated samples can also help the decoder to 'enlarge' the 'understandable' range to turn non-convex latent space to convex.

We use quantitative experiments to prove it. Firstly, we train both GSL-AE and DEAE on the Fonts-V2 dataset. We test the models on a test set in which the background colors and fonts colors are new colors unseen in V2. While testing, we feed the models with a pair of images from V2 and the test set that have the same attributes except for fonts colors and background colors. Then we compute the MSE loss between the images' latent spaces on the content, size and style part. The result shows that DEAE model has a lower average MSE loss of 71.2641 among the testing pairs while the GSL-AE model has a loss of 682237.8125. The latent codes of content, size and style remain similar even the colors of fonts and background are unseen while training, which means our DEAE model has a better ability for attribute disentanglement on unseen samples.

### 3.3 PRECISE CONTROL THE SYNTHESIS DIRECTION

Here we explore the distribution of disentangled representation and mining the relationship between movement in high dimension $x$ space and low dimension $z$ space to answer two questions: (1) How can we change the attribute value towards a specific value (e.g., change its background color to red color)? (2) Which direction of movement can help us to find new attributes?

As for the first question, for each background color, we train a binary color classifier to label interpolated points in the $z$ space and assign a color score for each of them, then we use SVM to find the boundary and obtain UDV for this attribute value. Since the UDV is the most effective direction to change the semantic score of samples, if we move $z$ value of the given image towards UDV, its related semantic score would increase fast. As is shown in Fig. 7 (a), the $1^{st}$ row represents the start image and the target image. In $2^{nd}$ row, if we move the $z$ value step by step, the proportion of red in the background color will increase obviously. The $3^{rd}$ row illustrates that for a small intra-class variance attribute, we can directly move the $z$ value to the cluster center of the target attribute, and this method guarantees that we can get the exact same background color.

As for exploring more new attributes, the combination of UDVs may be a good choice. for example, if the given picture is green, the new colors may fall in the path from green to blue as well as the path from green to red. Thus, it is reasonable to set our move direction as $v = v_{blue} + v_{red} - v_{green}$ ($v$ represents UDV). The $1^{st}$ row of Fig. 7 (b) shows the results of changing $z$ value with the combine vector $v_{blue} + v_{red} - v_{green}$. On the contrast, the $2^{nd}$ row only use $v_{blue}$ and the $3^{rd}$ row only use $v_{red}$. We can find that both the $2^{nd}$ and the $3^{rd}$ row only find one color while the $1^{st}$ row finds more.

### 3.4 DOWNSTREAM TASKS PERFORMANCE

We design a letter image classification experiment to explore the performance of DEAE as a generative model to improve the classifier model performance based on the downstream training dataset.

We tailored three datasets from Fonts, each of them has ten letters as labels (Table. 1). The large training set ($D_L$) and testing set ($D_{test}$) have the same number of images with the same attribute values of font, size and font color ,and different background colors. We take a subset of $D_L$ to form a small training set $D_S$ with fewer attribute values. We first train the resnet18 model for classification tasks on $D_L$ and $D_S$, calculating the test accuracy on the test dataset $D_{test}$. To evaluate the controllable synthesis performance of DEAE, we first train the model on $D_S$ and then we use the trained DEAE to synthesize 1000 more images. We combine the synthesized images with $D_S$ and form an augmented $D_{S+DEAE}$ training set. We also use the same training settings for GSL and form an augmented $D_{S+GSL-AE}$ training set. We compare two synthesized datasets by training classifiers with the same settings and calculating the corresponding testing accuracy (Table. 1). From $D_S$ to $D_{S+DEAE}$, the test accuracy increases from 71% to 76%, which shows the addition of synthesized

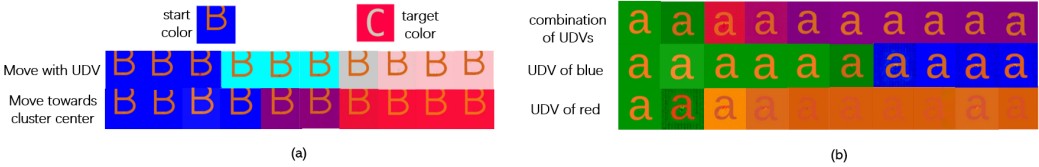

Figure 7: Changing to target attribute value and mining new attribute values

images from DEAE trained model to the small training set can improve the performance of the classification task. The accuracy gap from $D_{S+DEAE}$ to $D_{S+GSL-AE}$ shows that DEAE performs better than GSL-AE.

Table 1: Controllable augmentation performance (the $\star$ means that synthesized images with new attributes are added into training set)

| Attributes
Dataset | Letter | Size | Font Color | Back-Color | Fonts | Dataset Size | Training Accuracy | Test Accuracy |
|---|---|---|---|---|---|---|---|---|
| $D_L$ | 10 | 3 | 6 | 3 | 10 | 5400 | 98% | 94% |
| $D_S$ | 10 | 2 | 3 | 3 | 3 | 540 | 99% | 71% |
| $D_{S+GSL-AE}$ | 10 | 2$\star$ | 3$\star$ | 3$\star$ | 3$\star$ | 540+1000 | 99% | 74% |
| $D_{S+DEAE}$ | 10 | 2$\star$ | 3$\star$ | 3$\star$ | 3$\star$ | 540+1000 | 99% | 76% |
| $D_{test}$ | 10 | 3 | 6 | 3 | 10 | 5400 | N/A | N/A |

## 3.5 DATASET BIAS ELIMINATION

Dataset bias may influence on the performance of model greatly. Mehrabi et al. (2019) listed lots of bias resources and proved that eliminate bias is significant. Our DEAE could be a solution to biased problem. We design experiments to demonstrate how to achieve fairness with DEAE.

We design three datasets, a biased training dataset $\mathcal{D}^B$, an unbiased training dataset $\mathcal{D}^{UB}$ and an unbiased test dataset $\mathcal{D}^T$. In $\mathcal{D}^B$, we entangle the two attributes letter and background color as dataset bias. $\mathcal{D}^B$ consists of three-part: G1, G2, and G3. The details of those datasets can be found in Table. 2, and the number of colors represents the number of background colors for each letter.

Then, we use $\mathcal{D}^B$ and $\mathcal{D}^{UB}$ to train letter classifier with resnet-18 respectively and test on $\mathcal{D}^T$. As is shown in Table. 2, the classifier trained on $\mathcal{D}^B$, only gets 81% test accuracy while classifier trained on $\mathcal{D}^{UB}$ obtains 99% test accuracy. One possible explanation is that the entanglement of background colors and letters in the biased dataset results in the bad performance of the biased model. Grad-Cam (Selvaraju et al. (2019)) is a method that can highlight the region of pictures which plays a decisive role in the classifying process. As shown in Fig. 8, the results of Grad-Cam proved that the classifier would regard background color as essential information if it entangled with letters. We use the perfect disentangled latent space of DEAE to solve the entangled bias in $\mathcal{D}^B$. We first train a DEAE use $\mathcal{D}^B$. then we train the letter classifiers on the latent space instead of image space, where we explicitly drop the background color related dimensions and use the rest latent code as input. After training, the accuracy of biased model in the test dataset is 98%. Hence, we eliminate the dataset bias with our DEAE model.

Table 2: Dataset setting and experiment results

| Dataset | | Number of letters | Number of colors |
|---|---|---|---|
| $\mathcal{D}^B$ | G1 | 15 | 1 |
| | G2 | 15 | 3 |
| | G3 | 22 | 6 |
| $\mathcal{D}^{UB}$ | | 52 | 6 |
| $\mathcal{D}^T$ | | 52 | 6 |

| Model
Dataset | resnet18 -bias | resnet18 -unbias | DEAE -bias |
|---|---|---|---|
| Test(Letters in G1) | 52.73% | 99.17% | 96.77% |
| Test(Letters in G2) | 82.63% | 98.67% | 98.97% |
| Test(Letters in G3) | 99.13% | 98.30% | 98.46% |
| Train | 99.44% | 98.82% | 99.98% |
| Test | **81.32%** | 98.63% | **98.11%** |

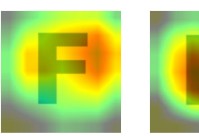
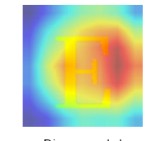
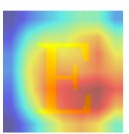

Bias model     Un–bias model     Bias model     Un–bias model

Figure 8: The influence of bias shown by Grad-Cam

## 4 CONCLUSION

We proposed a new kind of generative autoencoder: Disentangled Exploration Autoencoder (DEAE), which can achieve controllable synthesis by freely interpolating in disentangled latent space. DEAE tries to turn the non-convex latent space to convex for each attribute by 'reusing' the encoder to regularize the latent space of synthesized images. We show that DEAE outperforms state-of-the-art methods on attribute controllable synthesis tasks. We demonstrate how DEAE achieves precise latent space movement and novel attribute mining in perfect disentangled space by combining unit direction vectors. We also demonstrate that DEAE, as a generative model can improve the performance of downstream classification tasks as well as eliminate dataset bias, which provides a new solution for fairness problems.

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
