# OpenReview forum: "Generative Auto-Encoder: Controllable Synthesis with Disentangled Exploration"
_ICLR.cc/2021/Conference — Reject_

### Official Review · AnonReviewer2 · 2020-10-23
**Interesting problem but novelty and evaluation are limited**

**Rating:** 4
**Confidence:** 5

**Review:**

Summary:

The paper presents an approach for controllable generation with auto-ecoders. It uses interpolation in a disentangled latent space using Group Supervised Learning. Generated samples are encoded back to the latent space as additional regularization to improve sample quality. The authors perform experiments on Fonts and RaFD datasets for controllable generation. They also show applications of their model in improving performance of classifiers and reducing bias.

###########################################################

Strengths:

The problem of controllable image generation is important and current approaches are far from perfect. Autoencoders with disentangled latent space are a promising candidate for this problem as they can both reconstruct the input and control the attributes.

###########################################################

Weaknesses:
1. Literature review is inadequate and there are limited comparisons with related work. There are several other approaches that are not mentioned in the paper. Fader networks [A] also uses an encoder-decoder architecture trained to reconstruct images by disentangling salient information of the image and the values of attributes directly in the latent space. They provide a more comprehensive evaluation on CelebA and Flowers datasets while this paper use relatively simpler datasets, Fonts and RaFD. The authors need to clarify differences of their approach with [A] and provide comparisons with them. They need to include quantitative scores for their controllable generation. There are also other relevant papers such as Deep Feature Interpolation [B].
2. Experiments are mostly performed on the Fonts dataset which is relatively simple. It would be better to perform experiments on datasets such as CelebA which has attribute annotations.
3. Novelty of the proposed approach is limited as the authors use an existing method (GSL) for disentangling the latent space and also encoding the image back to latent space is proposed in other works such as InfoGAN and BicycleGAN.
4. To obtain UDVs the authors train a binary classifier for each color value which is not efficient.
5. It seems the bias experiment assumes that we know which attributes are entangled which might not be realistic.
6. In figure 7(a) changes of colors are discrete while continuous changes are more desirable.
7. In the results in Figure 2, it seems changing one attribute also affects the other.

############################################################

Reason for Rating:

Overall, while autoencoders with disentangled latent space are a promising candidate for controllable generation, experimental results and novelty of the proposed approach are inadequate for publication.

############################################################

References:

[A] Fader Networks: Manipulating Images by Sliding Attributes, Lample et al., NIPS 2017

[B] Deep Feature Interpolation for Image Content Changes, Upchurch et al., CVPR 2017

#############################################################

After author response: I thank the authors for their answers. However, as noted by other reviewers, experimental results and comparisons with related work are lacking, and I cannot increase my score without major changes to the paper.

---

> ### Author Response · Authors · 2020-11-25
> **Reviewer2 Response - Thanks for your feedback**
>
> We want to thank the reviewer for their time spent and useful comments that will help us improve this paper.
>
> Some additional details on the paper:
>
> 1. We will investigate more related works and compare these works with our model. The main difference between DEAE and Fader network is that we don’t need an extra discriminator to examine whether an attribute has been changed when training. Instead, we reuse the Encoder to guarantee the rest attributes are not changed while leaving the desired attribute free of inspection and we have an advantage compared with GAN based method: GAN training should carefully design and select the hyperparameters of G and D, while can not guarantee the convergence of training.
> 2. We will test our model on CelebA and Flowers datasets as suggested. But the Fonts dataset is still complex to some extent, with 52 letters, 10 colors (foreground and background), 3 sizes, and 100 font styles, producing 1,404,000 different combinations. This is enough to illustrate the distinction between our model and others, especially in attribute controllable synthesis tasks.
> 3. Similar to AnonReviewer3 Cons 3.
> 4. Because we use the decision boundary of an SVM to compute UDV. Since SVM is designed for binary classification problems, the most prevailing strategy for an SVM to solve multi-class classification problems is one-versus-rest. Maybe we can try different methods to derive UDV, but using SVM is the most direct method that follows intuition.
>
> 5. We do not need to know which attribute is entangled with the letter because after DEAE creates a disentangled latent space, we can use only the letter related dimensions as input for letter classification which has similar results.
>
> 6. Since we didn’t add loss in the training process constraining the change rate of the value of latent space and that of the corresponding attribute to be the same, a small change in latent space value might cause a huge change in the corresponding attribute or vice versa. Therefore, when we interpolate the latent space continuously, the attribute is not guaranteed to change continuously.
> 7. Figure 2 is confusing. We will reorganize this figure. The ‘original’ row presents all possible background colors. There is no vertical relationship among the three lines: ‘original’, ‘GSL-AE’, and ‘DEAE’, which means the ‘original’ row does not represent the input of the next two rows. All images are randomly picked and we want to show controllable generation for new background and font colors by interpolation in latent space while keeping the unmodified attribute as is.

---

### Official Review · AnonReviewer4 · 2020-10-26
**This paper proposed to refine an encoding by latent code interpolation followed by decoding/reencoding for cycle reconstruction. A further exploration approach UDV is proposed to generate unseen combination of attributes.**

**Rating:** 3
**Confidence:** 5

**Review:**

Summary

This paper proposed to explore a given disentangled latent code and explore it through random interpolations. The idea is to ensure these interpolated codes rely inside data convex hull by enforcing latent code reconstruction by first decoding and then reencoding this explored codes. The authors proposed a way to manipulate the final latent space by using Unit Direction Vector (UDV) to generate unseen attribute values. The results are illustrated on 2 datasets (Fonts and RaFD) both qualitatively and quantitatively.

Reason for score

Overall I vote for reject, this paper has limited contribution and lack of consistency (disconnected ideas: DEAE and UDV), clarity (respective contributions of GSL-AE and DEAE) and justification (spurious and not convincing experimentations).


Pros
1. The general topic of having more controlable synthesis by having more disentangled representation if of clear interest for the field.
2. The idea of reconstructing random interpolations of valid latent codes by decoding/reencoding is interesting. I don't think this is novel, but not widspread enough in the field so far, so the interest remains.

Cons
1. While the topic is of interest for the field, the interest of the proposed approach is clearly limited since it relies on a preprocessing step (GSL-AE) of which it is already the role. A clearer distinction of the contributions of each step would help to illustrate the additional interest of DEAE. Unfortunatly the paper relies too much on GSL-AE which limits the potential scope of their main contribution which could be wider.
2. The title is misleading since it presents the approach to proposed a "disentangled exploration" while it is essentially provided by GSL-AE used as a preprocessing step. Thus, it would also be good to mention that the approach is supervised (like GSL-AE).
3. The paper lacks of clarity: UDV or "dataset bias elimination" seems to be  spurious addition and are not really well connected in the paper. It's hard to understand how it contributes to illustrate the interest of DEAE.
4. Results are not convincing.
- MSE loss between latent spaces from different methods cannot be compared at all. We can imagine spurious differences only due to scale differences for instance.
- Fig 2. is mentioned p.5 to illustrate the qualitative superiority of DEAE over GSL-AE: not really obvious of this state.
- Section 3.3 + fig 7.: interpolation is not smooth which contradicts the quality of the representation and the interest of UDV.
- Section 3.4: the idea of evaluating the performance of generative methods through their capacity of improving a classification task by augmenting the initial dataset, is very indirect and not really convincing.
- Gradcam results (Fig. 8) does not illustrate any improvement in favor of background information for the "unbiased model"
5. "Dataset bias elimination"
- This section seems spurious. Nothing in the proposed approach is really specific to such an application appart the global context to which it is attached. I don't see what this section brings to the paper.
- The way the unbias model is obtained is not realistic since you need to know what information has to be dropped, whereas in real applications you generally don't know the source of bias.
6. While having thinner way of exploring the latent space is interesting, the proposed UDV approach sounds spurious and ad hoc and does not seem to be connected to DEAE.

Questions
1. What is the context of application of your approaches since it relies on a supervised approach to create the initial disentangled embedding ? Coutrolable synthesis is generally understood in an unsupervised setting.
2. Is there a link between UDV exploration and the way DEAE is learnt ?
3. Why using "random interpolation" for inference and illustrating performance (in Fig. 5) ? It would be clearer with standard interpolation between left and right bounds. I also would expect that interpolations would reach (or at least get closer to) the targeted right bound.
4. How do you ensure that MSE (from section 3.2) can be compared between totally different latent spaces ?
5. Section 3.4: Why D_S is used rather than D_L for further analyses ? (to get augmented datasets)


Minor comments
1. Intro: Flows could be mentionned on top of VAE and GANs
2. Intro on GAN & VAE limits are too much like a caricature regarding recent results. VAE can be used to generate HD images now, and GAN are much easier to train these days with GP strategy or different learning rates between generator and discriminator for instance. Approaches like InfoGAN could be mentioned to illustrate standard strategies to control GANs.
3. Fig 2. Not clear which kind of interpolation has been used for GSL-AE and DEAE since neither font, font size, font color, letter nor background color are interpolated in the figures. This figure would be more powerful to stick to one (or a few) letter(s) and change only the considered attribute (background or font color). Here the comparison and the illustration of the effect is not easy.
4. Section 2. "whic h maps"  "which maps"
5. notation "*" has not been introduced (like in f_theta^* and g_phi^*)
6. L_reg has not been introduced or referred in the paper.
7. Fig 7. Choosing a fg color different from the targeted bg color would be better

---

> ### Author Response · Authors · 2020-11-25
> **Regarding the reviewer's cons**
>
> We want to thank the reviewer for their time spent and useful comments that will help us improve this paper.
> Cons:
> 1. For the contribution of DEAE please refer to the part for all reviewers.
> 2. The method for exploration in latent space can be used in different latent spaces such as GANs, and the preprocessing is not necessary to be a supervised approach. Please refer to comments for all reviewers for the contribution of DEAE.
> 3. DEAE creates a disentangled, convex latent space and makes the latent space more robust through the interpolation. UDV is designed for explaining the latent space and exploring the distribution for each disentangled attribute-related latent space created by DEAE. To some extent, DEAE is the foundation of UDV and UDV provides more insight to understand the performance of DEAE, such as gain more explainable attribute values like new colors unseen in the training set.
> The dataset bias elimination is a downstream task of the DEAE. DEAE can help disentangle the latent space, which can help to eliminate the bias in the dataset since the entangled attributes will be disentangled in latent space. Please refer to section 3.5 for more details.
> 4.
> (1) We are working on conducting more experiments on the quantitative comparison.
> (2) When we use interpolation to controllably change the background color, the results from GSL shows that other attribute values also changed: foreground letter merged with background and the boundary has artifacts. The results of DEAE shows a clear controllable synthesis with no artifacts on other attributes out of background color.
> (3) The interpolation results of colors are not guaranteed to be continuous in the latent space since we didn’t have a loss function to constrain the change rate of the latent space, a tiny change while interpolation may cause a huge change in the attribute.
> (4) The evaluation of generative methods can be conducted in many aspects and the augmentation of biased datasets is a downstream task of the generative method. A classifier trained using a biased dataset can usually result in bad performance on the test set. DEAE can generate more instances with new attribute values which can aid the problem for the biased dataset.
> (5) The Gradcam result shows that when deciding the output of the classification, the biased model focuses on the wrong areas like background information, while the unbiased model focuses on the character itself. Therefore the DEAE model can help train an unbiased model that has a more meaningful ‘attention’ on the input data.
> 5. (1) The dataset bias elimination is a downstream task of the DEAE. DEAE can help disentangle the latent space, which can help to eliminate the bias in the dataset since the entangled attributes will be disentangled in latent space. This experiment shows the quality of disentanglement through a downstream task.
> (2) Yes, the source of bias, in reality, is not known most of the time. But in our experiment, it doesn’t matter whether we know about the source of the bias since all attributes will be disentangled in the latent space. The information about the source of bias in our manually created dataset is not used for training the classifiers.
> 6. Please refer to comment 3.

---

> ### Author Response · Authors · 2020-11-25
> **Regarding the reviewer's questions and minor comments**
>
> We want to thank the reviewer for their time spent and useful comments that will help us improve this paper.
>
> Questions:
> 1. As we mentioned in the comments to all reviewers, the DEAE approach does not necessarily depend on the supervised approach, it can be applied to any method with a latent space like GANs. We are working on more experiments on other models.
> 2. Please refer to comment 3. DEAE is the foundation of exploration using UDV.
> 3. Random interpolation is used because there are multiple dimensions in that latent space. There is no guarantee that different dimensions in latent space are co-related so that normal interpolation won't give good results in finding new attribute values.
> 4. The MSE loss needs normalization, we are working on conducting more experiments on the quantitative comparison.
> 5. D_L serves the role of baseline. We evaluate the ability for helping downstream tasks by creating an augmented training set based on D_S. It turns out that under the same setting, the augmented set generated by DEAE has the best performance.
>
> Minor comments:
> 1. We will have more related works cited in the introduction including Flow-based methods in the final version.
> 2. Yes GANs are easier to train and VAEs can have HD generation results, and we will have more cited works like InfoGAN in the paper. What we want to do is to have more insight into the area of attribute controllable synthesis. Our method has advantages like it’s easy to train, and it’s more like a module that can be used to models like GANs and VAEs. Our approach now does have many drawbacks but we will improve it in the future.
> 3. Figure 2 is confusing. We retitled this figure. The ‘original’ row only presents all possible background colors. There is no vertical relationship among the three lines: ‘original’, ‘GSL-AE’, and ‘DEAE’, which means the ‘original’ row does not represent the input of the next two rows. All images are randomly picked and we only want to illustrate that DEAE generates images with less noise and more clarity compared with GSL-AE.
> 4. Thank you for pointing out the typo.
> 5. f_^theta is the notation of the encoder in the model while g_^phi is the decoder
> 6. We will add an explanation and equation of the L_reg.
> 7. We will try to use another foreground color. Actually, the target background color in the figure is red instead of orange, the exploration along UDV of red gives the result of new colors like orange.

---

### Official Review · AnonReviewer1 · 2020-10-27
**Good paper however the results can to be more exhaustive**

**Rating:** 5
**Confidence:** 4

**Review:**

Summary: The paper proposes a new framework for adding a generative component to autoencoders without using adversarial or variational techniques. Further, the true motivation is to overcome issues due to naive interpolation (resulting in out of distribution samples) and to provide a semantic controllable synthesis in autoencoders. Empirical results show the proposed DEAE succeeds in both.

Strengths:
* I appreciate the intuition and examples throughout the paper, regarding the method itself and description of the experiments and desired outputs. This makes the paper easy to read and follow.
* The provided results support the claims (although they lack exhaustiveness, see below).


Weaknesses:
* Although I agree with the intuition of the method, no theoretical analysis is provided to justify the proposed solution.
The empirical results seem to confirm the statements of contribution but in a very limited setup. A more exhaustive presentation would be more convincing.

Although the paper builds on previous work, I find it quite interesting and significant to have a method for controllable synthesis. If the authors address my concerns, I am willing to increase my score.

Questions:

Methodology:
The main novelty in the proposed method is the L_{reg} loss term in the training of the DEAE, which was never explicitly written or explained beyond the intuitive interpretations.


Is there any theoretical direction you can propose for justifying the claim that adding the regularisation loss helps to turn non-convex to convex latent spaces? If not, at least a toy experiment to validate this assumption might help.

Missing reference:
The technique proposed for novel attribute mining, UVD is quite similar to the concept vectors proposed in TCAV [1].

Experiments:
Only a few quantitative results were presented, and those are also missing standard deviations. Please add those to make the results complete.

The results for experiment 3.2, were the MSE losses normalized? If the latent spaces correspond to different models, how do we know if the numbers are comparable?

Other baselines:
- Generative models: at least some comparison to vanilla VAE, GANs as a sanity check (if the claim is generative AE)
- Why is GSL-AE not used in 3.4?
- Could you please provide more plots of the nice results (Figures 5 and 6) in the appendix just to confirm the presented results were not “cherry-picked”.


Minor:
- In Figure 5 the acronym for general AE is ‘AE’ instead of ‘GAE’
- I don’t see the need for  ‘non-adversarial’ in the title.

[1] Kim, Been, et al. "Interpretability beyond feature attribution: Quantitative testing with concept activation vectors (tcav)." International conference on machine learning. PMLR, 2018

---

> ### Author Response · Authors · 2020-11-25
> **Regarding the reviewer's concern**
>
> We want to thank the reviewer for their time spent and useful comments that will help us improve this paper.
>
> As for Weaknesses:
> Yes, the theoretical proof is hard but we give the explanation in 2.2.
>
> As for Questions:
> 1.  We now have a formal description of the regularization reconstruction loss in the updated main paper (Eq.1 and 2)
> 2.  Yes, as we mentioned in the main paper 2.2, adding the regularization will help enrich the latent space that the decoder can ’understand’, and the perfect disentanglement regularization enforce the latent representation of every attribute is understandable by the decoder, i.e., it forces, in the limit of infinite interpolated samples, the disentangled latent representation of every attribute to be convex. we have several experiments to qualitatively demonstrate that:
> As shown in Fig.5, when doing controllable synthesis on background, size, and fonts, compared with GSL-AE, DEAE can synthesize high-quality images that change only the background-color attribute while keeping the other attributes unmodified.
> 3.  We will cite this paper in the final version. Thanks!
> 4.  We will add the standard deviations in the final version. Thanks!
> 5.  Other baselines:
> (1) we are running comparison experiments for more VAE and GAN based methods for comparison.
> (2) We use GSL-AE and show the results in Table. 1
> (3) Yes, we will add more results in the appendix of the final version.
>
> As for minor:
> 1. Thank you for pointing out, GAE may look like generative AE or other AEs which will be confusing. What we used is a simple autoencoder.
> 2. We changed our title regarding the part of non-adversarial.

---

### Official Review · AnonReviewer3 · 2020-10-28
**Need to clearly state scope, method, contributions and to provide a more rigorous experimental setup.**

**Rating:** 3
**Confidence:** 4

**Review:**

# Summary
This paper proposes a new method called Disentangled Exploration Auto-Encoder (DEAE). This new method is based on “(Ge at al.) Zero-shot synthesis with group-supervised learning” to which a modified cyclic loss term is added. This method is trained on datasets with label supervision.

# Pros
1. Compared to “(Ge at al.) Zero-shot synthesis with group-supervised learning” the results seem to be more visually pleasing.

# Cons
1. The method is not clearly presented. In particular the loss terms are not mathematically expressed as equations. I don’t know if we’re expected to read the paper this method is based on to have detailed equations. In any case, the added cyclic loss is expressed as an equation either, so it’s really hard to say what the method really does. It’s also unspecified how the latent space is allocated to attributes.
2. The scope of the method is not clearly presented either: I was under the impression that the proposed method was comparable to other unsupervised auto-encoders while in fact it requires label supervision
3. It seems the paper claims to be more than it actually is: if I understand correctly, the sole contribution of this paper is a cyclic loss term, compared the (Ge et al) paper which can be seen as a regularizer.
4. Experiments are mostly focused on visual inspection of images and do not appear impressive to me (except for the toy dataset of colored letters, but then I don’t know what the SotA is like). Very few numerical results (save for dataset bias elimination) are presented. Results are not compared to other SotA methods, except the (Ge at al) paper which the method is based upon. Experimental setup is very incomplete.

# Questions and nits
1. It would have benefited my comprehension to mention in the abstract that the proposed method requires attribute/label supervision. In its current form, it seems to claim to solve disentanglement for unsupervised auto-encoders and I find it misleading.
2. The mention of the generative ability without GAN based training in the abstract is also confusing: the proposed method seems orthogonal to GANs and could be combined with them.
3. Several times I see the term “perfect disentanglement” to describe the improved disentanglement that this method offers compared to (Ge et al). What I don’t understand is what makes it “perfect”, can’t it be further improved?
4. “<mention unsupervised auto-encoders>. We propose a different solution to empower precise attribute controllable synthesis ability on autoencoders: DEAE”. To be fair, it’s also a solution to a different problem scope. The proposed method uses attributes while the cited methods don’t. So it’s really a solution to a different problem altogether.
5. Typo “whic h” => “which”
6. “Fig. 4 (d) shows that we can combine the UDVs to dicover new attribute values.” Aside from the typo on "discover", it’s unclear how you discover new attributes values (which I assume are centroid like the example you mentioned before for the blue color). Here instead, my understanding is that UDV just provides a vector along which values of interest may lie. It seems the eventual decision to make a value an attribute is manually decided by a human after inspecting the effects along an UDV axe.
7. Downstream task performance. It is a toy dataset and it’s hard to really tell the real power of the proposed method. A lot of information is missing, what are the sizes the $D_S$ and $D_L$? The only reported numbers are $D_S$ vs $D_{S+DEAE}$. What is the unreported accuracy gap between $D_{S+DEAE}$ and $D_{S+GSL-AE}$ that leads to the later conclusion that DEAE performs better? What is the accuracy for $D_L$? No information is known about the classifier network architecture(s), parameter sizes, tuning and whether or not they overfit or what other causes could be responsible for the observed results.

=====POST-REBUTTAL COMMENTS========
I thank the authors for the response and the efforts in the updated draft. Some of my queries were clarified, particularly concerning missing experimental details. However, unfortunately, I still think more needs to be clarified in the actual paper write up, notably on the points of non-adversarial as well as the method description.

---

> ### Author Response · Authors · 2020-11-25
> **Reviewer3 Response - Thanks for your feedback**
>
> We want to thank the reviewer for their time spent and useful comments that will help us improve this paper.
>
> Some additional details on the paper:
> Cons:
> 1. Equation: We have updated the detailed equation of the regularization reconstruction loss as Eq. 1 and 2 in the updated paper.
> 2. ~ 4. As we mentioned above (to all reviewers), the preprocess, GSL, does need the attribute label of data, however, the DEAE itself is an unsupervised manner to constrain the latent space which achieves controllable synthesis.
> Our method can further increase the controllable synthesis performance given a disentangled latent space (GSL), so most of our experiments compared with our baseline model, GSL-AE. There may be no clear controllable synthesis baselines that are similar to our settings. But we are trying to use other synthesis methods, GAN-based controllable synthesis which also needs the attribute label during training, to conduct attribute controllable synthesis and compare with our DEAE.
>
> Questions and nits:
> 1. See comments to all reviewers.
> 2. Yes, as we mentioned above, DEAE can be compatible with GAN based methods by regularizing the latent space with disentangled interpolation. However, GANs intrinsically have the generative ability while auto-encoders do not. DEAE can empower the attribute controllable synthesis ability to an autoencoder based method.
> 3. The term “perfect disentanglement property” (defined in Sec. 2.2) is an ideal property for the latent space where we can achieve attribute controllable synthesis by freely interpolating. DEAE creates a positive loop where the disentangled representation and exploration can help each other, which helps create a perfect disentangled autoencoder.
> 4. See comments to all reviewers.
> 5. We have corrected this typo and all other grammar errors we find.
> 6. Yes, the combination of UDVs provides directions to find new attribute values with high probability. We can log the synthesized image along the path and find the new attribute value. We can also use the latent distance between synthesized images and know images in disentangled latent space to distinguish the new attribute value. Basically, UDVs provide a method to explore the distribution of each attribute value in a disentangled latent space.
> 7. All dataset and accuracy information is reported in Table 1. We use Resnet-18 as our classifier, we will provide more training details in the final version.

---

### Author Response · Authors · 2020-11-25
**Equation of loss function, contribution and novelty**

We want to thank all the reviewers for their time spent and useful comments that will help us improve this paper!

1 Contribution and novelty compared with baseline GSL.

Our proposed DEAE is designed to achieve attribute controllable synthesis based on disentangled exploration. However, the results we showed are based on a preliminary disentangled latent space which is created by GSL (need attribute label during training). To achieve attribute controllable synthesis, DEAE trying to create a positive loop between disentanglement and interpolation which achieve two things: (1) trying to create a ‘perfect disentanglement’ latent space (Sec. 2.2) (2) turn the ’non-convex’ latent space for each attribute to be convex (Sec. 2.2).  Theoretically, our DEAE can be used on different latent spaces, we are trying to conduct more experiments on the latent space of VAE and GAN to show the improvement of controllable synthesis.

2 We have updated the detailed equation of the regularization reconstruction loss as Eq. 1 and 2 in the updated paper.

---

### Decision · Program_Chairs · 2021-01-07
**Final Decision**

**Decision:**

Reject

**Comment:**

Overall, the paper makes some interesting and intuitive observations regarding the autoencoders with a cycle consistency, and aims at achieving controllable synthesis via a disentangled representation. However, the overall consensus was that the manuscript needs further iterations:

In particular:
The ideas should be made more precise using mathematical arguments, as it stands some ideas are (e.g. DEAE and UDV) disconnected.

The scope needs to be clarified, e.g. respective contributions of GSL-AE and DEAE, use of label information

More numerical/quantitative evaluations, the current experimentation is not convincing enough, needed for better justification (spurious and not convincing experimentations)

The English of the manuscript could be improved as it occasionally hampers the flow.